# Nradd Acts as a Negative Feedback Regulator of Wnt/β-Catenin Signaling and Promotes Apoptosis

**DOI:** 10.3390/biom11010100

**Published:** 2021-01-14

**Authors:** Ozgun Ozalp, Ozge Cark, Yagmur Azbazdar, Betul Haykir, Gokhan Cucun, Ismail Kucukaylak, Gozde Alkan-Yesilyurt, Erdinc Sezgin, Gunes Ozhan

**Affiliations:** 1Izmir Biomedicine and Genome Center (IBG), Dokuz Eylul University Health Campus, Inciralti-Balcova, 35340 Izmir, Turkey; ozgun.ozalp@msfr.ibg.edu.tr (O.O.); ozge.cark@msfr.ibg.edu.tr (O.C.); yagmur.azbazdar@msfr.ibg.edu.tr (Y.A.); betuel.haykir@uzh.ch (B.H.); goekhan.cucun@biochemie.uni-freiburg.de (G.C.); ikuecue1@uni-koeln.de (I.K.); gozde.alkanyesilyurt@ibg.edu.tr (G.A.-Y.); 2Izmir International Biomedicine and Genome Institute (IBG-Izmir), Dokuz Eylul University, Inciralti-Balcova, 35340 Izmir, Turkey; 3Institute of Physiology, Switzerland and National Center of Competence in Research NCCR Kidney, University of Zurich, CH-8057 Zurich, Switzerland; 4Institute of Zoology-Developmental Biology, University of Cologne, 50674 Cologne, Germany; 5Science for Life Laboratory, Department of Women’s and Children’s Health, Karolinska Institutet, 17177 Stockholm, Sweden; erdinc.sezgin@ki.se; 6MRC Weatherall Institute of Molecular Medicine, MRC Human Immunology Unit, University of Oxford, Oxford OX39DS, UK

**Keywords:** Nradd, p75 neurotrophin receptor, Wnt/β-catenin signaling, apoptosis, death receptor

## Abstract

Wnt/β-catenin signaling controls many biological processes for the generation and sustainability of proper tissue size, organization and function during development and homeostasis. Consequently, mutations in the Wnt pathway components and modulators cause diseases, including genetic disorders and cancers. Targeted treatment of pathway-associated diseases entails detailed understanding of the regulatory mechanisms that fine-tune Wnt signaling. Here, we identify the neurotrophin receptor-associated death domain (Nradd), a homolog of p75 neurotrophin receptor (p75^NTR^), as a negative regulator of Wnt/β-catenin signaling in zebrafish embryos and in mammalian cells. Nradd significantly suppresses Wnt8-mediated patterning of the mesoderm and neuroectoderm during zebrafish gastrulation. Nradd is localized at the plasma membrane, physically interacts with the Wnt receptor complex and enhances apoptosis in cooperation with Wnt/β-catenin signaling. Our functional analyses indicate that the N-glycosylated N-terminus and the death domain-containing C-terminus regions are necessary for both the inhibition of Wnt signaling and apoptosis. Finally, Nradd can induce apoptosis in mammalian cells. Thus, Nradd regulates cell death as a modifier of Wnt/β-catenin signaling during development.

## 1. Introduction

Wnt/β-catenin signaling is an evolutionarily conserved signaling pathway that controls a plethora of cellular processes including proliferation, migration, apoptosis, cell fate determination and tissue patterning during development and adult homeostasis [1,2,3]. Due to these essential roles, misregulation of the signaling pathway has been associated with various human diseases, including cancer [1,4,5,6,7]. Pathway misregulation in cancer is triggered via either mutational alterations, such as gain-of-function mutations in β-catenin and Tcf transcription factors and loss-of-function mutations in the β-catenin destruction complex components, or nonmutational alterations such as silencing of extracellular Wnt antagonists through epigenetic mechanisms [8,9,10,11].

Wnt/β-catenin signaling is thought to operate with its core components as follows: If there exist no active Wnt ligands in the environment, the so-called Wnt-off state, the β-catenin destruction complex containing Axin, adenomatous polyposis coli (Apc), glycogen synthase kinase 3 (Gsk3) and casein kinase 1 (Ck1) bind to β-catenin and leads to its proteasomal degradation after phosphorylation [12,13]. The cell switches to the Wnt-on state when the Wnt ligand triggers the formation of a complex between itself, the receptor Frizzled (Fz) and the co-receptor low-density lipoprotein receptor-related protein (Lrp) 5/6 at the plasma membrane. This interaction appears to recruit cytoplasmic proteins Disheveled and Axin to the (co)receptors at the membrane. Further receptor clustering in turn causes Lrp6 phosphorylation and subsequent internalization of the receptor complex by endocytosis [14,15]. The removal of Axin from the destruction complex, direct inhibition of Gsk3 kinase activity by the phosphorylated cytoplasmic portion of Lrp6 and sequestration of Gsk3 from the cytoplasm into multivesicular bodies are collectively assumed to prevent the degradation of β-catenin [16,17,18]. β-catenin then enters the nucleus and interacts with the transcription factors of the T cell factor (Tcf)/lymphoid enhancer factor (Lef) family to regulate the expression of its target genes [19].

Due to the deleterious consequences of aberrant signal transduction, it is not surprising that Wnt/β-catenin signaling is controlled by a large number of positive and negative regulators. Tight regulation of the pathway at the plasma membrane is achieved by a complex network of extracellular or membrane-bound modulators [14,20,21]. Many of these pathway modulators, including Axin2, Dkk1, Waif1/5T4 and Lypd6, are also Wnt targets and act in feedback regulation, a key feature of Wnt pathway regulation [22,23,24,25,26]. Since these modulators are usually not required for vital cellular activities and are highly specialized for the signaling pathway, they constitute a worthwhile option for targeted therapeutic interventions that can dampen abnormal Wnt pathway activity.

Here, we identify the *neurotrophin receptor-associated death domain* (*nradd*) gene, encoding for a single-pass transmembrane protein with a C-terminal death domain, as a transcriptional target of Wnt/β-catenin signaling during the embryonic development of zebrafish. Nradd exhibits a high degree of homology with the p75 neurotrophin receptor (p75^NTR^), which has been characterized as a receptor for members of the neurotrophin family [27,28]. The neurotrophins are a family of growth factors that are essential for growth, differentiation, survival, apoptosis and regeneration of the neurons in vertebrates [29,30]. There are four types of neurotrophins in mammals: nerve growth factor (NGF), brain-derived neurotrophic factor (BDNF), neurotrophin-3 (NT-3) and neurotrophin 4 (NT-4) [31]. Neurotrophins bind to and signal through two main types of receptors: tropomyosin receptor kinase (Trk) family receptors, also known as neurotrophic tyrosine kinase receptors, and p75^NTR^, a member of the tumor necrosis factor receptor superfamily (TNFRSF) [32,33,34]. Being produced as larger precursors called proneurotrophins that turn into mature forms via proteolytic cleavage, neurotrophins exhibit distinct binding affinities for their receptors [29,35]. Proneurotrophins, for example, show high affinity to a complex composed of p75^NTR^ and sortilin [36,37].

Neurotrophins induce neuronal survival or cell death by activating Trk receptors and p75^NTR^, respectively [38]. The binding of mature neurotrophins with Trk results in the activation of phosphoinositide 3-kinase (PI3K), phospholipase Cγ1 (PLCγ1) and mitogen-activated protein kinase (MAPK) pathways and promotes cell growth, differentiation and survival [39]. On the other hand, mature neurotrophins that bind to p75^NTR^ can enhance neurotrophin binding to Trk receptors and Trk signal transduction through protein kinase B (PKB or AKT) and MAPKs, depending on the circumstances. They can either enhance survival by the nuclear factor-κB (NF-κB) pathway or antagonize the action of Trk through the activation of the JUN N-terminal kinase (JNK) and Rhoa pathways [39,40,41,42,43,44]. This ultimately generates a paradoxical paradigm that harbors a matter of life and death depending on Trk receptor and p75^NTR^ signaling [45]. Mouse NRADD has been found to induce apoptosis in neuroblastoma cell lines through the activation of caspase 8 and independently of the mitochondrial pathway, or so-called intrinsic apoptosis [27]. However, little is known about its mechanism of action and potential interaction with a signaling pathway.

We describe Nradd as a feedback inhibitor of Wnt/β-catenin signaling during zebrafish development and in mammalian cells. Nradd localizes to the plasma membrane and interacts with the Fz8 and Lrp6 receptors. Our functional analyses show that Nradd acts together with Wnt/β-catenin signaling to promote apoptosis during development and that both the death domain and N-glycosylated N-terminus are essential for its apoptotic function. Finally, Nradd can also enhance apoptosis in human embryonic kidney and neuroblastoma cell lines. Overall, Nradd acts as a negative modulator of Wnt/β-catenin signaling and a key player in the regulation of apoptosis during development.

## 2. Materials and Methods

### 2.1. Transgenic Fish Lines

Transgenic zebrafish (*Danio rerio*) lines Tg(hsp70l:wnt8a-GFP)^w34^ and Tg(hsp70l:Mmu.Axin1-YFP)^w35^ were outcrossed to wild type (wt) AB zebrafish [22,46]. The embryos were heat shocked and identified as described previously [22]. The Tg(7XTcf-Xla.Siam:nlsmCherry)^ia^ transgenic line was used as a reporter of Wnt/β-catenin signaling, outcrossed to wt zebrafish and embryos were sorted as described previously [47]. Animal experiments were approved by the Animal Experiments Local Ethics Committee of Izmir International Biomedicine and Genome Institute (IBG-AELEC) on 26.07.2017 with the protocol number 12/2017.

### 2.2. Cloning

RNA was isolated from whole embryos at 24 h post-fertilization (hpf) using a Direct-Zol RNA kit (Zymo Research, Irvine, CA, USA) and cDNA was synthesized with an iScript reverse transcription kit (Biorad, Hercules, CA, USA) using a 1:1 mixture of oligo (dT) and random primers. iScript reverse transcriptase (RT) was replaced with water for (-RT) controls. The following products were amplified with the corresponding primer pairs using 1 µL of cDNA. Zebrafish wt Nradd-EGFP: forward primer 5′-AAAAAGGATCCCCACCATGAAAGGAGCAACTGAAGC-3′ and reverse primer 5′-AAAAAGAATTCGACCACTGATACTCCTTGCG-3′; Nradd death domain deleted (DDD)-EGFP: forward primer 5′-AAAAAGGATCCCCACCATGAAAGGAGCAACTGAAGC-3′ and reverse primer 5′-AAAGAATTCATCCTGTTTGCTGTCCCGTTTA-3′. Both PCR products were digested with BamHI and EcoRI and ligated into a pCS2P+ vector that has EGFP. The Nradd DDD construct was generated through the deletion of a 95 amino acid region between the 318th and 413th amino acids at the C-terminal region. Nradd glycosylation site deleted (GSD)-EGFP was generated by site-directed mutagenesis at wt Nradd-GFP using overlap extension PCR. The Nradd GSD construct was generated through the deletion of a 101 amino acid region between the 22nd and 123rd amino acids at the N-terminal region. The first round of PCR was conducted with the two primer pairs forward1 5′-AAAAAGGATCCCCACCATGAAAGGAGCAACTGAAGC-3′, reverse1 5′-ACACTCCCCATCCTGGCCCACAGCCAAAGGCCATCTT-3′ and forward2 5′- AAGATGGCCTTTGGCTGTGGGCCAGGATGGGGAGTGT-3′, reverse2 5′-AAAAAGAATTCGACCACTGATACTCCTTGCG-3′. The second round of PCR was performed by using the purified PCR product of the first round as the template and the primers forward1 and reverse2. The purified PCR product of the second round of PCR was again digested with BamHI and EcoRI and ligated into a pCS2P+ vector that has EGFP. For Fz8a-EGFP, zebrafish Fz8a was amplified with the forward primer 5′-AGAATTCAACCACCATGGAGTGCTACCT-3′ and the reverse primer 5′- GGATCCTCAGACTTGGGACAAAGGC-3′. The PCR product was digested with EcoRI and BamHI and ligated into a pCS2P+ vector that has EGFP. For Fz8a-mRuby3, mRuby was amplified with the forward primer 5′-AAGGCGCGCCTATGGTGTCTAAGG-3′ and the reverse primer 5′- AATCTAGATTACTTGTACAGCTCGTCCATGCCAC-3′ from the mRuby3 plasmid. The PCR product was digested with AscI and XbaI and ligated into an Fzd8a-EGFP plasmid after excising EGFP with the same restriction enzymes. For Nradd-mRuby3, mRuby3 was amplified with the forward primer 5′-AAATCTAGAATGGTGTCTAAGGGCGAAGA-3′ and the reverse primer 5′- AAAGATATCTTACTTGTACAGCTCGTCCAT-3′ from the mRuby3 plasmid. The PCR product was digested with XbaI and BsaBI ligated into a wt Nradd-EFGP plasmid after excising EGFP with the same restriction enzymes. Successful cloning was verified by sequencing, restriction digestion and agarose gel electrophoresis.

### 2.3. Capped Sense mRNA Synthesis, Microinjection and Whole-Mount in situ Hybridization (WMISH)

Capped sense RNAs of GFP, wt Nradd, Nradd-GFP, Nradd DDD-EGFP and Nradd GSD-EGFP were synthesized using an mMessage mMachine Kit (Thermo Fisher Scientific, Waltham, MA, USA). mRNAs were injected into 1-cell zebrafish embryos as 250 pg for *nradd* and 20 pg for *Wnt8*. Embryos were fixed at indicated stages (30% epiboly, 50% epiboly, 60% epiboly, 80% epiboly, 100% epiboly, 3-somite, 5-somite, 10-somite or 24 hpf) in 4% paraformaldehyde (PFA) overnight. WMISH was performed with *nradd*, *mCherry*, *goosecoid, otx2, hoxb1b, foxg1a, her5* and *krox20* antisense RNA probes as described previously [48].

### 2.4. Quantitative PCR (qPCR)

cDNA was synthesized from RNA using a ProtoScript II First Strand cDNA Synthesis Kit (New England BioLabs, Ipswich, MA, USA) according to the manufacturer’s instructions. Zebrafish *rpl13a* or *β-actin* were used as the reference genes for normalization to determine the relative gene expression levels. qPCR was performed in triplicate using GoTaq qPCR Master Mix (Promega, Madison, WI, USA) in an Applied Biosystems 7500 Fast Real Time PCR machine (Foster City, CA, USA). The data were analyzed using GraphPad Prism 8 software (Graphpad Software Inc., San Diego, CA, USA). The values are mean ± standard deviation (SD) of three samples. The following primers were used: *mCherry* forward 5′-GAACGGCCACGAGTTCGAGA-3′ and reverse 5′-CTTGGAGCCGTACATGAACTGAGG-3′, zebrafish *sp5l* forward 5′-GCTTCACGCAGGTGTGGAT-3′ and reverse 5′-TTCTGGAGATGAGCTGGGAGT-3′, zebrafish *axin2* forward 5’-TAGTTTTGCCCCTGCCACG-3’ and reverse 5’-TCCCAGCTTGTAAGGAGGAATG-3’, zebrafish *cdx4* forward 5’-CCAGAGAAAATCAGAGCTGGCA-3’ and reverse 5’-TTGCACCGAGCCTCCACTATT-3’, zebrafish *nradd* forward 5’-AAGTGTCAGCCATGCCAAGA-3’ and reverse 5’-AGGAATACCGATGGGGCAATG-3’, zebrafish *rpl13a* forward 5′-TCTGGAGGACTGTAAGAGGTATGC-3′ and reverse 5′-AGACGCACAATCTTGAGAGCAG-3′, zebrafish *β-actin* forward 5′-GAAGGAGATCACCTCTCTTGCTC-3′ and reverse 5′-GTTCTGTTTAGAAGCACTTCCTGTG-3′.

### 2.5. Cell Culture

Human embryonic kidney 293T (HEK293T), neuroblastoma SH-SY5Y and bone osteosarcoma U2OS cell lines were purchased from ATCC (Manassas, VA, USA). HEK293T and U2OS cells were cultured in high-glucose Dulbecco’s modified Eagle’s medium (DMEM) and SH-SY5Y cells in DMEM F-12 at 37 °C in a 5% (v/v) CO_2_ humidified environment. All media were supplemented with 10% fetal bovine serum (FBS) and 1% penicillin–streptomycin.

### 2.6. Transfection and Luciferase Assay

HEK293T were seeded on 24-well plates and transfected in triplicate with 20 ng of Wnt8, with and without 300 ng of Nradd. SH-SY5Y cells were seeded on 24-well plates and transfected in triplicate with 250 ng of Nradd and stimulated with Wnt3a conditioned media (CM) for 6h. All cells were co-transfected with 20 ng of firefly luciferase reporter pGL3 BAR [49] and 5 ng of renilla luciferase reporter pGL4.73 hRLuc/SV40 (Promega, Madison, WI, USA) using Fugene HD Transfection Reagent (1 µg/1 µL, Promega, Madison, WI, USA). Twenty-four hours after transfection, reporter activity was measured with the dual luciferase reporter assay kit (Promega, Madison, WI, USA) in a Varioskan Flash multimode reader (Thermo Fisher Scientific, Waltham, MA, USA). Statistical analysis was performed using a Student’s *t*-test. Error bars represent SD, where * and ** indicate *p* < 0.05 and *p* < 0.01, respectively.

### 2.7. Subcellular Localization

Zebrafish embryos were injected with 250 pg Nradd-EGFP and 250 pg RFP-GPI at the 1-cell stage, dechorionated at 3 hpf, mounted in 3% methylcellulose, and imaged using a Zeiss LSM 880 confocal microscope (Carl Zeiss AG, Oberkochen, Germany). U2OS cells were seeded in a 6-well plate on a 35 mm glass-bottom dish with a 10 mm micro-well #1.5 cover glass (0.16–0.19 mm) and transfected with Fz8a-EGFP and Nradd-mRuby3 plasmids (500 ng each) using Lipofectamine 3000 reagent at 24 h. The next day, cell membranes were stained with CellMask Deep Red at a concentration of 5 µg/mL for 5 min. The cells were imaged in Leibovitz’s L-15 medium without phenol red using a Zeiss LSM 780 confocal microscope.

### 2.8. Isolation of Giant Plasma Membrane Vesicles (GPMVs)

U2OS cells were seeded in a 6-well plate on a 35 mm glass-bottom dish with a 10 mm micro-well #1.5 cover glass and transfected with 500 ng Nradd-EGFP using Lipofectamine 3000 reagent at 48 h. At ~70% confluence, the cells were washed twice with GPMV buffer (10 mM HEPES, 150 mM NaCl, 2 mM CaCl_2_, pH 7.4) and incubated for 1 h at 37⁰C with GPMV buffer including 2 mM *N*-ethyl maleimide (NEM). After 1 h, the supernatant containing vesicles was collected, incubated with Atto647N-labeled 1,2-dipalmitoyl-sn-glycero-3-phosphoethanolamine (Atto647N-PE) for membrane staining and imaged using a Zeiss LSM 780 confocal microscope.

### 2.9. Co-Immunoprecipitation and Western Blotting

HEK293T cells were seeded in 6-well plates and transfected with 500 ng Nradd-GFP and 500 ng LRP6-HA. At 48 h, Wnt induction was performed for 6 h and cells were washed with ice-cold PBS and lysed with NOP buffer (10 mM HEPES KOH pH 7.4, 150 mM NaCl, 2 mM EDTA, 10% glycerol, 1% NP40 (Igepal CA-630, Sigma-Aldrich, St. Louis, MO, USA). Lysate was centrifuged at 300 g for 5 min at 4 °C. Supernatant was precipitated with a SureBeads™ Protein G Magnetic Beads Co-Immunoprecipitation Kit (BioRad, Hercules, CA, USA) according to the kit protocol. For western blotting, samples were dissolved in 5X loading dye and separated by SDS gel electrophoresis on a 10% acrylamide-bis acrylamide gel. Proteins were transferred from the gel to a polyvinylidene fluoride (PVDF) membrane (GE Healthcare Life Science, Chicago, IL, USA). Membranes were blocked in 5% milk powder for 45 min at room temperature (RT) and incubated with the following antibodies at the corresponding dilutions. Primary antibodies: rabbit anti-GFP ((D5.1) XP, 1:1000; Cell Signaling Technology, Danvers, MA, USA) and mouse anti-HA (1: 2000; OriGene Technologies, Rockville, MD, USA). Secondary antibodies: anti-rabbit IgG, horseradish peroxidase (HRP)-linked and anti-mouse IgG, HRP-linked (both 1:2500; Cell Signaling Technology, Danvers, MA, USA).

### 2.10. Fluorescence Cross-Correlation Spectroscopy (FCCS)

The molecular interaction between Nradd protein and the components of the Wnt–receptor complex was examined by comparing the diffusion characters on the membrane using a fluorescence cross-correlation spectroscopy (FCCS) technique. FCCS measurements were performed on a Zeiss LSM 780 confocal microscope equipped with 63X oil immersion objective (1.4 NA). GFP- and mCherry-labeled proteins were excited with 488 and 561 nm lasers, respectively. Detection intervals of 500–550 nm and 600–690 nm were used to detect green and red signals, respectively. Curves were analyzed using the FoCuS-point software package [50]. Cross-correlation percentages were calculated by the amplitudes of autocorrelation and cross-correlation curves of molecules tagged with two different fluorophores at the plasma membrane. Calculations were performed by using the GNO values obtained from the FoCuS-point program. Among the values of GFP- and mCherry-labeled molecules, the one which was closer to the cross-correlation GNO value was considered for comparisons. This value was normalized using the positive control. The experiment was performed in U2OS cells that were seeded on 0.17 mm cover slides. The next day, cells were transfected with 500 ng of each plasmid using Lipofectamine 3000 reagent. The diffusion rates were measured at 24 h. The following combinations of molecules were used: (1) Fz8a-EGFP + Lrp6-mCherry, (2) Fz8a-EGFP + Lrp6-mCherry + Wnt3a CM, (3) Fz8a-mRuby3 + Nradd-EGFP, (4) Fz8a-mRuby3 + Nradd-EGFP + Wnt3a CM, (5) Lrp6-mCherry + Nradd-EGFP, (6) Lrp6-mCherry + Nradd-EGFP + Wnt3a CM. All values were normalized to the positive control EGFP-mCherry. GPI-EGFP + Fzd8a-mRuby3 + Wnt CM was used as a negative control. Wnt3a CM was produced from murine L cells. Cells were grown in a 10 cm plate in DMEM supplemented with 10% FBS. At 90% confluence, cells were diluted 1:10 and seeded in new 10 cm plates. Wnt3a CM was collected at 2 days, 4 days and 6 days, and added to the cells 20 min prior to measurement.

### 2.11. Immunofluorescence Staining

Zebrafish embryos produced by crossing wt AB and Tg(*hsp70l:*wnt8a-GFP)^w34^ were injected with the capped sense RNAs of control GFP (100 pg), wt Nradd-GFP (250 pg), Nradd-DDD (250 pg) or Nradd-GSD (250 pg) at the 1-cell stage. Injected embryos were heat shocked at 60% epiboly, dechorionated at 24 hpf by adding 100 μL of 10 mg/mL pronase in 10 mL E3 medium (5 mM NaCl, 0.17 mM KCl, 0.33 mM CaCl_2_, 0.33 mM MgSO_4_) and fixed in 4% PFA in PBST (1× PBS, 0.1% Tween-20) overnight at 4 °C. Embryos were incubated in ice-cold methanol at -20 °C for 24 h. Methanol was removed and embryos were washed twice with 1 mL 1× PDT (1× PBST, 0.3% Triton-X, 1% DMSO) for 30 min by rocking gently at RT. Next, embryos were incubated with 500 μL blocking buffer (1× PBST, 10% heat-inactivated fetal bovine serum, 2% bovine serum albumin) at RT for 1 h and incubated with rabbit anti-cleaved caspase 3 antibody (1:400; Cell Signaling Technology, Danvers, MA, USA) at 4 °C overnight. The next day, embryos were washed twice with 1× PDT, incubated with rhodamine (TRITC)-AffiniPure donkey anti-rabbit IgG (1:400; Jackson ImmunoResearch Laboratories, Inc., West Grove, PA, USA) for 2 h at RT, re-washed with 1× PDT, mounted in 50% glycerol in PBS and imaged using a Zeiss LSM 880 confocal microscope.

### 2.12. Annexin V-FITC Apoptosis Assay

HEK293T or SH-SY5Y cells were seeded in 6-well plates. For the control and Nradd, cells were transfected with firefly luciferase reporter pGL3 BAR (100 ng) and the plasmid containing Nradd without GFP tags (250 ng), respectively, according to the manufacturer’s protocol. For the positive controls, cells were treated with 0.3 mM H_2_O_2_ and 1µM staurosporine for 18 h. Relevant cells were treated with Wnt3a CM overnight to induce Wnt signaling. Twenty-four hours after transfection, supernatants and trypsinized cells were collected. Cells were washed and stained with annexin V-FITC and 7AAD using an FITC Annexin V Apoptosis Detection Kit with 7-AAD (BioLegend, San Diego, CA, USA). Flow cytometry was performed using a BD Fortessa and data were analyzed with FlowJo 8.8.6 software (Tree Star Inc., Ashland, OR, USA). Experiments were performed in three biological replicates and data are representative of at least three independent experiments.

## 3. Results

### 3.1. Nradd is Transcriptionally Regulated by Wnt/β-Catenin Signaling during Development

To identify new modulators of Wnt/β-catenin signaling, we exploited the fact that various pathway components are also transcriptional targets of the pathway. To identify Wnt targets, we conditionally activated or inhibited Wnt/β-catenin signaling by giving transgenic zebrafish embryos a single heat shock at three independent developmental stages, namely, gastrula, somitogenesis and organogenesis, and performed RNA sequencing. By screening for genes that are upregulated upon activation and downregulated upon inhibition, i.e., positively regulated by the pathway, during all three developmental stages, we identified neurotrophin receptor-associated death domain (*nradd*) that has no previously defined function in Wnt/β-catenin signaling and in vertebrate embryonic development. At late gastrula stages, *nradd* was expressed at the marginal zone and later became concentrated at the posterior sites of the embryo (Figure 1A). During organogenesis, its expression was downregulated in the posterior regions and observed at a weaker level throughout the body (Figure 1A). To confirm the gene expression profiling results, we investigated *nradd* expression levels at gastrula, somitogenesis and organogenesis by in situ hybridization in heat shock-inducible transgenic zebrafish embryos Tg(hsp70l:Mmu.Axin1-YFP)w35 (hs:Axin1) and Tg(hsp70l:wnt8a-GFP)w34 (hs:Wnt8) by overexpressing pathway inhibitor Axin1 and pathway activator Wnt8, respectively. In all three developmental stages, the inhibition of Wnt/β-catenin signaling significantly suppressed *nradd* expression, while its activation upregulated *nradd* (Figure 1A). To support these data quantitatively, we performed qPCR in the same conditions and found that the expression of both *nradd* and the direct Wnt/β-catenin target gene *sp5l* increased with activated Wnt signaling in hs:Wnt8 embryos and decreased with inhibited signaling in hs:Axin1 embryos at all developmental stages tested (Figure 1B–D). These results confirm that *nradd* is a transcriptional target of Wnt/β-catenin signaling and its expression is broadly regulated by the pathway during development.

### 3.2. Nradd Acts as an Inhibitor of Wnt/β-Catenin Signaling

To test whether *nradd* has an influence on Wnt/β-catenin signaling, we first cloned wild-type zebrafish *nradd* into a pCS2P+ expression vector with or without EGFP, termed Nradd and Nradd-EGFP, respectively. Next, we overexpressed *nradd* mRNA in a transgenic reporter of Tcf/Lef-mediated transcription, Tg(7XTcf-Xla.Siam:nlsmCherry)^ia^ (7xTcf:mCherry). Reporter expression dramatically decreased in *nradd*-overexpressing embryos at late gastrula, somitogenesis and organogenesis stages (Figure 2A). Activation of Wnt8-mediated β-catenin signaling generates a range of phenotypes from mild neuroectodermal posteriorization to loss of forebrain and notochord in zebrafish embryos (classes 2 to 4, Figure 2B) [23,51]. To test whether *nradd* overexpression can rescue these phenotypes, we activated Wnt/β-catenin signaling in hs:Wnt8 transgenic embryos by giving a heat shock at early gastrula (shield stage), which generated class 3 and 4 phenotypes at 24 hpf (Figure 2C). *nradd* overexpression indeed significantly restored the phenotypes to class 2 and 3 (Figure 2C). Moreover, *nradd* significantly downregulated the expression of *axin2*, *cdx4* and *sp5l*, three direct Wnt/β-catenin target genes in these embryos [25,46,52]. Strikingly, *nradd* overexpression was also able to reduce the expression of these genes in wild-type embryos (Figure 2D–F). Finally, zebrafish *nradd* efficiently suppressed Wnt/β-catenin signaling in HEK293T and SH-SY5Y cells, as evidenced by the activation of the firefly luciferase pBAR reporter of Tcf/Lef-mediated transcription (Figure 2G–H) [49]. These results strongly suggest that *nradd* acts as a negative feedback regulator of Wnt/β-catenin signaling during zebrafish development and can suppress Wnt signaling in mammalian cells.

### 3.3. Nradd Suppresses Wnt-Mediated Patterning of the Mesoderm and the Neuroectoderm

During gastrulation, wnt8-mediated β-catenin signaling has essential roles in the specification of the ventrolateral mesoderm and the repression of the dorsal organizer [51]. To test whether *nradd* affects mesodermal patterning of the embryo, we examined how the organizer reacts to *nradd* overexpression during early development. *nradd* caused a significant enlargement of the dorsal organizer region marked by goosecoid (gsc) expression [53], not at the late blastula (30% epiboly) stage, but at the early gastrula (60% epiboly) stage (Figure 3A,B). This suggests that *nradd* can relieve wnt8-mediated repression of the dorsal organizer. Wnt/β-catenin signaling is also required for the induction of posterior neural fates during gastrulation. *nradd* overexpression caused an extension of the early forebrain and midbrain region marked by otx2 [54] and a concomitant restriction of the posterior neuroectodermal region marked by hoxb1b [55] at late gastrula (100% epiboly, Figure 3C). To further investigate the role of *nradd* in later patterning of the neuroectodermal fates, we tested whether it can rescue wnt8-induced posteriorization of the zebrafish embryo brain. *nradd* overexpression rescued the anterior neuroectodermal fates, shown by the forebrain marker foxg1a, which completely disappeared in wnt8-overexpressing embryos during mid-somitogenesis (10–13 somites (Figure 3D). Thus, we conclude that *nradd* acts as a feedback inhibitor of wnt8-mediated patterning of mesoderm and neuroectoderm during zebrafish gastrulation.

### 3.4. Nradd Localizes to the Plasma Membrane and Interacts with the Wnt–Receptor Complex

Since *nradd* encodes for a transmembrane protein, we next aimed to examine its subcellular localization and test whether it interacts with the Wnt–receptor complex. Nradd-EGFP localized to the plasma membrane and co-localized with the membrane bound RFP-GPI (glycosylphosphatidylinositol) in zebrafish embryos during the early blastula stage (Figure 4A). When expressed in the human osteosarcoma epithelial cell line U2OS, we observed that Nradd tagged with mRuby partially localized to the plasma membrane along with the canonical Wnt receptor Fz8a tagged with EGFP (Figure 4B). Fractions of both proteins, Nradd to a larger extent, were detected in the intracellular space (mostly endoplasmic reticulum [ER]), presumably due to overexpression. To further confirm the plasma membrane localization of Nradd-EGFP, we prepared giant plasma membrane vesicles (GPMVs) from Nradd-EGFP-expressing U2OS cells. GPMVs are pure plasma membrane encapsulating soluble cytoplasmic content while being devoid of any organelles [56]. In GPMVs, Nradd-EGFP was clearly detectable at the surface of the vesicles, confirming its plasma membrane localization (Figure 4C) [56]. Next, we asked whether there is a physical interaction between Nradd and components of the Wnt–receptor complex at the plasma membrane. First, Nradd coimmunoprecipitated with the canonical Wnt co-receptor Lrp6, showing a potential interaction between these two proteins (Figure 4D). Second, we applied fluorescence cross-correlation spectroscopy (FCCS) that can be used to analyze the synchronous movement (co-diffusion) and binding behavior of two fluorescently labeled proteins with single molecule detection sensitivity in live cells [57]. Representative graphics for low cross-correlation (negative control), high cross-correlation (positive control and Lrp6+Nradd+Wnt3a) are provided (Appendix A). In addition, Wnt3 stimulation increased the interaction of all three combinations of molecules at the membrane, defined by percentages of cross-correlation (Figure 4E). Thus, Nradd appears to diffuse together with both the Fz8a receptor and Lrp6 co-receptor at the membrane and this diffusion is further enhanced by canonical Wnt stimulation. These results together show that Nradd localizes at the plasma membrane and physically interacts with the components of the Wnt–receptor complex.

### 3.5. Nradd Acts Together with Wnt/β-Catenin Signaling to Promote Apoptosis during Development

Although Nradd has been shown to promote apoptosis in neuroblastoma cell lines [27], its potential role in embryonic development has not been described. Nradd has four amino-terminal cysteine-rich domains (CRDs) conserved among TNFRSF members and located between the 23rd and 184th amino acids and a carboxy-terminal death domain located between the 326th and 404th amino acids. In addition to wt Nradd, we thus generated two other constructs, Nradd-DDD and Nradd-GSD, in which the C-terminal region including the death domain and the N-terminal region including the N-glycosylation site and the two CRD domains were deleted, respectively (Figure 5A). Initially, to address whether Nradd can induce apoptosis during development, we overexpressed wt *nradd* mRNA in zebrafish embryos and performed immunostaining for activated caspase 3, a marker of apoptosis. Since Nradd can suppress wnt8-mediated patterning of the neuroectoderm during development, we focused on the brain regions of embryos. Nradd was sufficient to significantly enhance apoptosis on its own and also acted together with wnt8-mediated β-catenin signaling, which is activated in hs:Wnt8 transgenic embryos by giving a heat shock at 60% epiboly (Figure 5B,C). In contrast to Nradd, Nradd-DDD and Nradd-GSD could neither inhibit canonical Wnt signaling in the 7xTcf:mCherry Wnt reporter line (Figure 5D) nor induce apoptosis in zebrafish embryos (Figure 5E,F). Thus, Nradd appears to promote apoptosis together with Wnt/β-catenin signaling and both the C-terminal death domain and the N-glycosylated N-terminal domain are essential for its apoptotic function.

### 3.6. Zebrafish Nradd Promotes Apoptosis in Human Embryonic and Neuroblastoma Cell Lines

The *nradd* gene has been inactivated by mutation and is nonfunctional in humans (NRADDP, Gene ID: 100129354). To test whether zebrafish Nradd is functional in humans, first we transfected human embryonic kidney 293T (HEK293T) cells with the zebrafish Nradd. Nradd significantly induced apoptosis but did not act together with Wnt3a (Figure 6A,B). Moreover, zebrafish Nradd was able to significantly enhance apoptosis in human neuroblastoma SH-SY5Y cells, again independently of Wnt3a (Figure 7A,B). These results suggest that while the apoptotic role of Nradd is conserved across vertebrates, its additive effect with Wnt signaling is not conserved.

## 4. Discussion

Owing to its essential roles in embryonic development and the maintenance of tissue homoeostasis, Wnt/β-catenin signaling is tightly regulated by various pathway modulators. Here, we characterize the functional role of Nradd, a poorly characterized homologue of p75^NTR^, in pathway regulation during zebrafish development. Our data suggest that Nradd i) is a Wnt/β-catenin target during zebrafish development, ii) can suppress Wnt/β-catenin signaling during development and in mammalian cells, iii) efficiently inhibits Wnt8-mediated ventralization of the mesoderm and posteriorization of the neuroectoderm, iv) localizes at the plasma membrane and physically interacts with the Wnt–receptor complex, v) promotes apoptosis by acting together with Wnt/β-catenin signaling during development and vi) can also enhance apoptosis in mammalian cells.

Neuronal survival and innervation rely on the presence of developing neuronal populations that receive accurate trophic factors from neighboring cells and activate intracellular signaling pathways [58]. Nevertheless, neuronal cell death exhibits an essential cell death program by which a neuron directs its own destruction and is essential for the proper development and functioning of the nervous system [59]. The death receptor, or so-called extrinsic apoptosis, pathway is one of the essential mechanisms of neuronal cell death [60]. The pathway is mediated by the interaction of death ligands with TNFRSF death receptors at the cell surface and further recruitment of Fas-associated death domain protein (FADD), which in turn leads to autoproteolytic cleavage and the activation of initiator caspases *and downstream executioner caspases including caspase 3 [61]. Aberrant regulation of the death receptor signaling pathway has been associated with neuronal death in* neurodevelopmental disorders, *neurodegenerative conditions such as traumatic brain injury, amyotrophic lateral sclerosis and stroke and psychiatric disorders [62,63]*. p75^NTR^*,* a pleiotropic signaling molecule, can act as a co-receptor for various receptors, including Trk, sortilin and Nogo, and mediate a variety of cellular functions from survival and axonal growth to apoptosis through several signaling pathways [29,64,65,66,67]. Our results unravel Wnt/β-catenin signaling as a novel interaction partner of Nradd, a p75^NTR^ homolog, in the regulation of apoptosis during embryonic development. Wnt signaling regulates the early and late stages of apoptosis during the development of various organs, including the brain, the limbs and the heart [68,69,70,71]. During neural development, the ability of Wnt signaling to facilitate or prevent apoptosis is highly dependent on the cellular context and other signaling pathways that integrate apoptosis with other cellular processes [72]. Wnt signaling *has been associated with*
*neurotrophin signaling* in the coordination of neuronal development and differentiation [73,74]. Wnt/β-catenin signaling has previously been shown to regulate expression of the death receptors DR6 and TROY in the brain epithelium to mediate blood–brain barrier development in the central nervous system [75]. An elegant work has recently shown that cells with abnormal Wnt/β-catenin activity are eliminated via apoptosis with the involvement of cadherin proteins and Smad signaling and this elimination is required for proper anterior–posterior patterning [76]. Our results support this finding that Wnt/β-catenin signaling can induce apoptosis during development and further unravel that Nradd can potently enhance the apoptosis-promoting role. Since Nradd is both a feedback inhibitor of Wnt/β-catenin signaling and can potently induce apoptosis in the zebrafish brain during development, it will be very interesting to further analyze which specific cells of the brain undergo apoptosis upon Nradd expression.

p75^NTR^ has been shown to promote the apoptosis of neurons and oligodendrocytes in development, regeneration and pathological conditions [67,77,78,79,80]. Several studies have unraveled the roles of p75^NTR^ structural domains in the activation of death receptor signaling. The extracellular domain of p75^NTR^ appears to be responsible for the conformational changes that propagate the signal to its death domain, which further recruits the interactors of various signaling pathways [81,82,83]. The death domain is also necessary for p75^NTR^-induced neuronal apoptosis [83,84]. The induction of apoptosis is dependent on the N-terminal domain that was shown to be modified by N-glycosylation, i.e., the attachment of an oligosaccharide to the protein in the ER [27]. Our functional analysis with the Nradd constructs has demonstrated that both the N-glycosylated N-terminal region and the death domain-containing C-terminal region are necessary for the suppression of Wnt/β-catenin signaling and the induction of apoptosis. Thus, we suggest that the N-terminal region undergoing N-glycosylation and the C-terminal death domain are key regions that can be exploited for the control of cell death mechanisms.

Mouse NRADD has been reported to induce cell death in primary neuronal cells and neuroblastoma cell lines [27,85,86]. Our data unravel the apoptotic role of zebrafish Nradd in both human embryonic and neuroblastoma cell lines and further suggest that Nradd induces early apoptosis characterized mainly by alterations occurring at the plasma membrane. Unlike encoding for a transmembrane protein in mice and zebrafish, the human *NRADD* gene carries inactivating mutations that have converted it into a pseudogene in humans. Strikingly, our data show that zebrafish Nradd can not only inhibit Wnt/β-catenin signaling but also promote apoptosis in human embryonic and cancer cell lines, indicating that Nradd has an evolutionarily conserved function in the regulation of Wnt signaling and apoptosis. It is possible that human p75NTR homologs might compensate for the absence of Nradd in humans. While Nradd appears to act together with Wnt signaling to enhance apoptosis during development, Wnt activation does not further enhance Nradd-mediated apoptosis in human cells. The activation of Wnt/β-catenin signaling could, however, sensitize human melanoma cells to apoptosis induced by the TNF-related apoptosis-inducing ligand (TRAIL/APO2L) [87]. In contrast, the secreted Wnt inhibitor Dkk3 has been found to induce Fas death receptor signaling in human ovarian cancer cells, suggesting a proapoptotic role for Dkk3 [88]. Furthermore, death receptor signaling can contrariwise regulate Wnt signaling. Several inhibitors of NGF and NGF receptors, such as Ro 08-2750 (targets NGF), K252a (targets TrkA) and LM11A-31 (targets P75), have been shown to increase β-catenin expression and cell migration in ovarian cancer cells [89]. Thus, we believe that the relationship between Nradd and Wnt signaling in the regulation of apoptosis mechanisms is context dependent. Future studies on understanding the relationship between Nradd and Wnt signaling regulation in different contexts, such as neurodevelopment, neurodegeneration and cancer, are very likely to constitute a promising approach for the development of disease-specific therapies.

The role of Wnt/β-catenin signaling in neural development starts at the early induction of the neural plate and continues throughout subsequent patterning of the neuroectoderm during CNS development [90]. Our results reveal Nradd as a new player in this process with a robust capacity to modify canonical Wnt-mediated posteriorization of the neuroectoderm. Moreover, the pathway is known to be actively involved in the modulation of the synaptic function to maintain the basal neural activity [91]. Consequently, alterations in Wnt signaling have been implicated in many neurodegenerative and neuro-oncological diseases, raising the importance of the specific pathway targeting for efficient treatment strategies [92,93,94,95,96,97]. Besides, the misexpression of pseudogenes has been linked to various human diseases, including several cancers [98,99,100,101,102,103]. For example, deletion of the PTENP1 pseudogene locus in melanoma enhances miRNA-mediated suppression of PTEN and tumor progression [104]. Overexpression of a PTENP1 transgene could restore its tumor suppressor activity and inhibit tumor growth [105]. Therefore, the ability of Nradd to modulate Wnt signaling in human cells offers a promising target for therapeutic interventions in neurological disorders.

## 5. Conclusions

Our study reveals the death receptor-encoding nradd, a homolog of p75NTR, as a Wnt target during development and an inhibitor of the Wnt/β-catenin signaling pathway in zebrafish embryos and mammalian cells. Nradd can potently suppress canonical Wnt-mediated phenotypes in mesodermal and neuroectodermal patterning during zebrafish gastrulation. By physically interacting with the Wnt–receptor complex at the plasma membrane, Nradd acts together with Wnt/β-catenin signaling to promote apoptosis during development. Moreover, human embryonic and neuroblastoma cells undergo apoptosis upon Nradd overexpression. Thus, by both fine-tuning pathway activation at the plasma membrane and regulating cell death, the Wnt modulator Nradd might serve as an attractive target for the discovery of therapeutic interventions against Wnt-related human diseases.

## Figures and Tables

**Figure 1 biomolecules-11-00100-f001:**
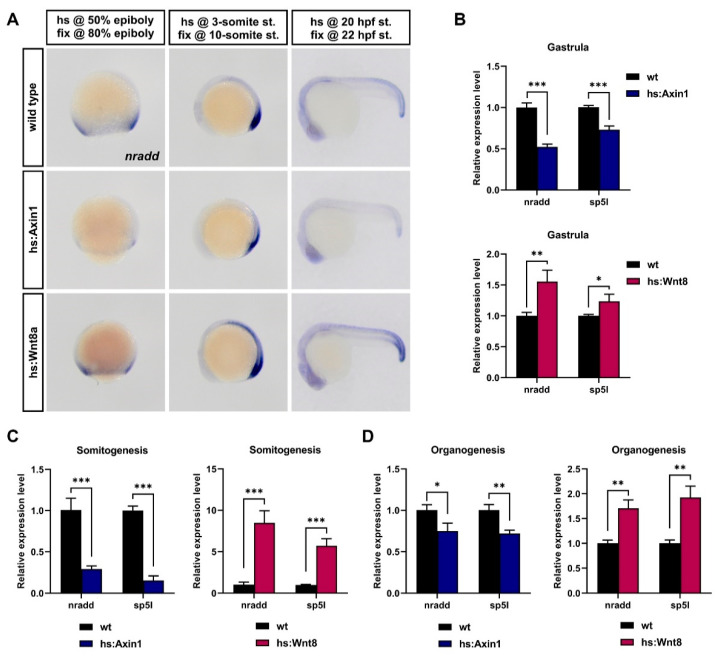
*nradd* is transcriptionally regulated by Wnt/β-catenin signaling during development. (**A**) Whole-mount in situ hybridization (WMISH) shows that *nradd* expression is downregulated in transgenic embryos expressing Axin1 (hs:Axin1) and upregulated in transgenic embryos expressing Wnt8a (hs:Wnt8a) at gastrula (80% epiboly), somitogenesis (10-somite) and organogenesis (22 hpf) stages. (n (80% epiboly): wild type 21/21 embryos, hs:Axin1 18/20 embryos, hs:Wnt8a 16/17; n (10-somite st.): wild type 24/24 embryos, hs:Axin1 24/24 embryos, hs:Wnt8a 28/30; n (22 hpf st.): wild type 19/19 embryos, hs:Axin1 23/25 embryos, hs:Wnt8a 19/20) hs: heat shock, fix: fixation, st: stage, hpf: hours post-fertilization. Three independent experiments were conducted. (**B**–**D**) *nradd* and *sp5l* expression levels determined by qPCR in hs:Axin1 and hs:Wnt8a transgenic embryos are shown relative to those in wild-type embryos at (**B**) gastrula, (**C**) somitogenesis and (**D**) organogenesis stages. Statistical significance was evaluated using an unpaired *t*-test. * *p* < 0.05, ** *p* < 0.01 and *** *p* < 0.001. Error bars represent ± standard deviation (SD, *n* = 3). Three independent experiments were conducted.

**Figure 2 biomolecules-11-00100-f002:**
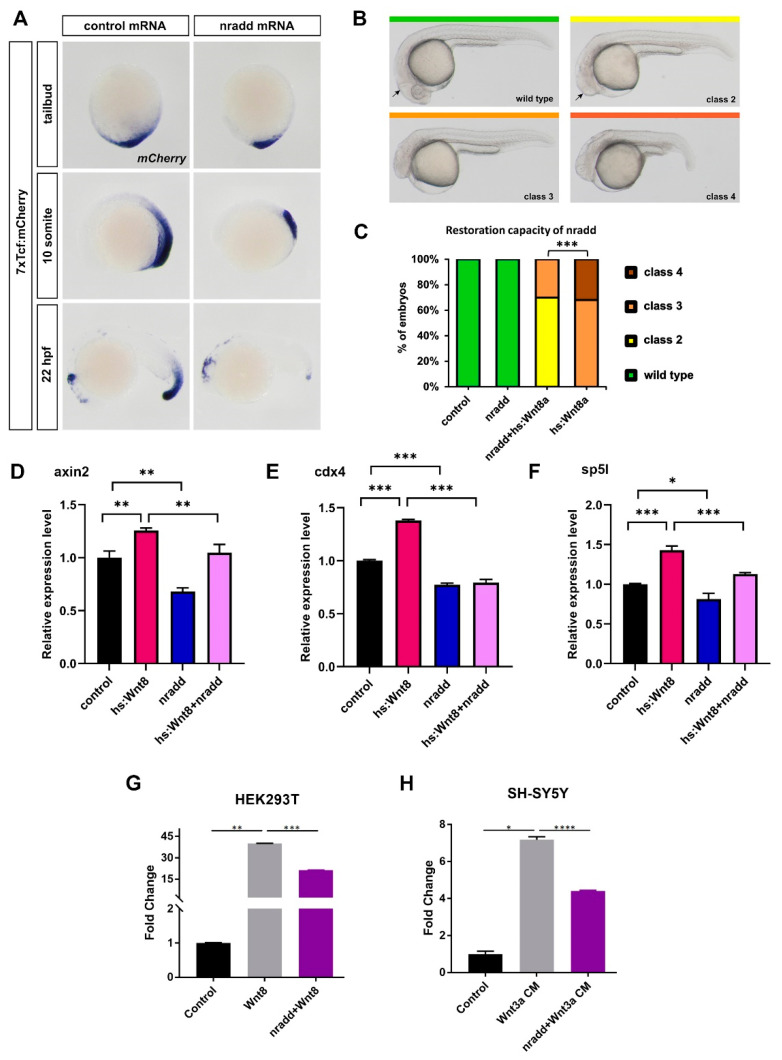
*nradd* acts as an inhibitor of Wnt/β-catenin signaling. (**A**) Wnt/β-catenin reporter activity is reduced in 7xTcf:mCherry embryos injected with 250 pg *nradd* mRNA. (n (tailbud): control 26/26 embryos, nradd 27/31 embryos; n (10-somite): control 26/26 embryos, nradd 25/26 embryos, n (22 hpf): control 21/21 embryos, nradd 29/30 embryos). Three independent experiments were conducted. (**B**) Classification of phenotypes at 24 hpf caused by *wnt8* overexpression in hs:Wnt8a transgenic embryos after 1 hour of heat shock during gastrulation. Class 2: no eyes, reduced forebrain, normal midbrain–hindbrain boundary (mhb, arrow), class 3: no forebrain, severely reduced midbrain, no mhb, class 4: abnormal notochord development. (**C**) Phenotypes in wild-type (wt) or hs:Wnt8 transgenic embryos injected with *nradd* mRNA (150 pg) or equimolar amounts of control mRNA heat shocked at shield stage during gastrulation and scored at 24 hpf. Nradd rescues Wnt8-induced phenotypes. (n (tailbud): control 53 embryos, nradd 46 embryos, nradd+hs:Wnt8a 56 embryos, hs:Wnt8a 45 embryos). Three independent experiments were conducted. (**D**–**F**) Expression levels of direct Wnt-target genes *axin2*, *cdx4* and *sp5l* determined by qPCR in *nradd*-overexpressing wt embryos (nradd) relative to those in wt embryos injected with *GFP* mRNA (control) and in *nradd*-overexpressing hs:Wnt8 embryos (hs:Wnt8+nradd) relative to those in hs:Wnt8 embryos injected with *GFP* mRNA (hs:Wnt8). All target genes are reduced by *nradd* in both wt and hs:Wnt8 embryos. Statistical significance was evaluated using an unpaired *t*-test. * *p* < 0.05, ** *p* < 0.01 and *** *p* < 0.001. Error bars represent ± standard deviation (SD, n = 3). Three independent experiments were conducted. (**G**–**H**) Average and SD of the mean (error bars) values of pBAR luciferase reporter activity monitoring Wnt/β-catenin signaling activity (normalized to renilla luciferase activity) in (**G**) HEK293T and (**H**) SH-SY5Y cells where Wnt/β-catenin signaling is activated by Wnt8 or Wnt3a conditioned media (CM) and transfected with Nradd. Nradd significantly inhibited Wnt signaling in both cell lines. Statistical significance was evaluated using an unpaired *t*-test. * *p* < 0.05, ** *p* < 0.01, *** *p* < 0.001 and **** *p* < 0.0001. Error bars represent SD. Three independent experiments were performed.

**Figure 3 biomolecules-11-00100-f003:**
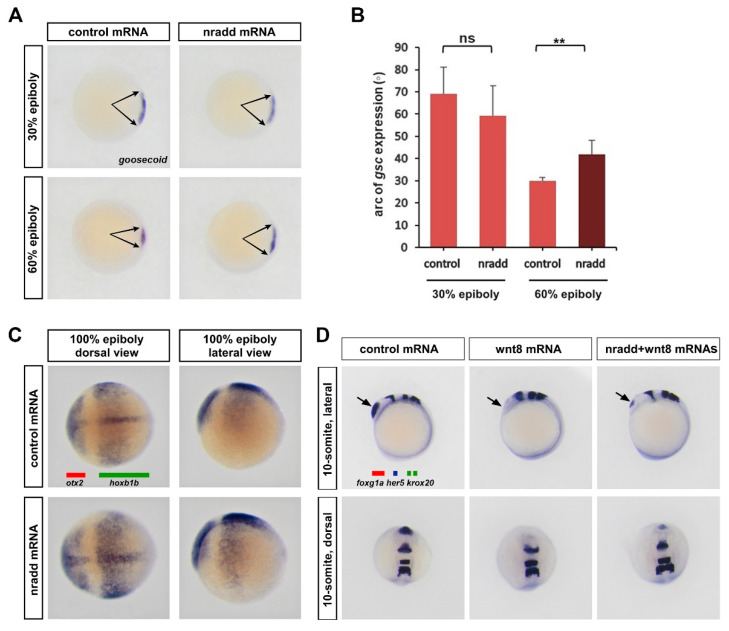
*nradd* suppresses Wnt-mediated patterning of the mesoderm and the neuroectoderm. (**A**) *nradd* mRNA (250 pg) causes expansion of the dorsal organizer domain (arrows) marked by *goosecoid* (gsc) WMISH, not at late blastula (30% epiboly, n: control 23/23 embryos, nradd 26/29 embryos), but at early gastrula (60% epiboly, n: control 27/27 embryos, nradd 32/33 embryos) stage zebrafish embryos. Three independent experiments were conducted. (**B**) Quantification of *gsc* expression shown in (A) by measurement of arc degree (29 embryos in control 30% epiboly, 33 embryos in nradd 30% epiboly, 35 embryos in control 60% epiboly, 37 embryos in nradd 60% epiboly). Error bars are SD. ** indicates *p* < 0.01 and ns is non-significant. (**C**) *nradd* mRNA (250 pg) results in expansion of the anterior neuroectodermal marker *otx2* (red bar) and a complementary reduction of the posterior neuroectodermal marker *hoxb1b* (green bar) defined by WMISH at the 100% epiboly stage (n: control 35/35 embryos, nradd 38/41 embryos). Three independent experiments were conducted. (**D**) *nradd* mRNA (250 pg) restores the telencephalon, marked by *foxg1a* (arrowhead), that is completely abolished by *wnt8* mRNA (20 pg). WMISH is performed to detect expression of three independent RNAs, the forebrain marker *foxg1a* (red bar), the midbrain–hindbrain boundary marker *her5* (blue bar) and the rhombomere 3/5 marker *krox20* (two green bars) in zebrafish embryos at the 10-somite stage (n: control 33/33 embryos, wnt8 35/36 embryos, nradd+wnt8 42/45 embryos). Three independent experiments were conducted.

**Figure 4 biomolecules-11-00100-f004:**
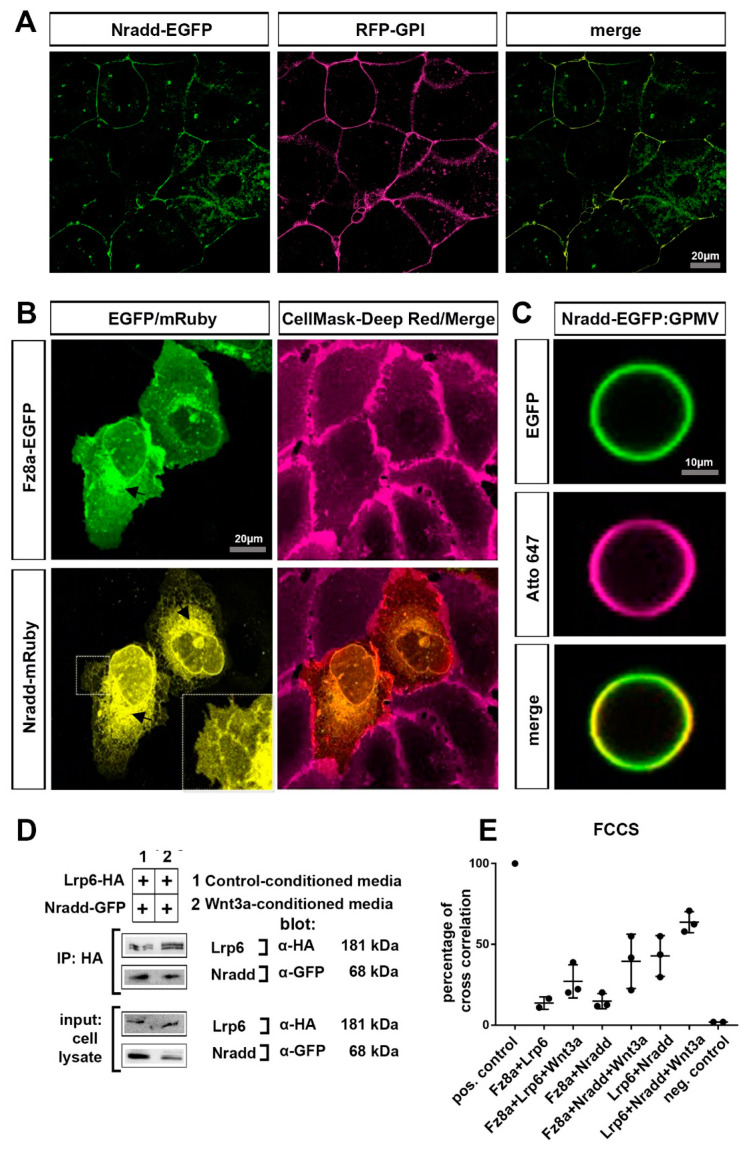
Nradd localizes to the plasma membrane and interacts with the Wnt–receptor complex (**A**) Nradd-EGFP (green) and RFP-GPI (red) localize at the plasma membrane of the enveloping layer cells of zebrafish embryos at 4 hpf (dome stage). (**B**) In U2OS cells, Fz8a-EGFP (green) co-localizes with the CellMask Deep Red (red) at the plasma membrane while Nradd-mRuby (yellow) localizes to the ER, but a significant fraction of it is in the plasma membrane, as seen from the zoom-in inset image. Arrows indicate protein transport from the ER to the plasma membrane. (**C**) Nradd-EGFP (green) localizes to the surface of cell-derived giant plasma membrane vesicles (GPMVs) and overlaps with the plasma membrane marker Atto647N-PE (red). (**D**) Nradd-GFP co-immunoprecipitates with Lrp6-HA in HEK293T cells. Three independent experiments were performed. (**E**) Fluorescence cross-correlation spectroscopy (FCCS) measurements show cross-correlation percentages between Fz8a-Lrp6, Fz8a-Nradd and Lrp6-Nradd, with/without Wnt3a stimulation at the plasma membrane. Nradd co-diffuses with Fz8a and Lrp6 at the membrane and this co-diffusion is enhanced by Wnt3 stimulation. EGFP and mCherry proteins were used as negative controls and their cross-correlation percentage is 1.70%.

**Figure 5 biomolecules-11-00100-f005:**
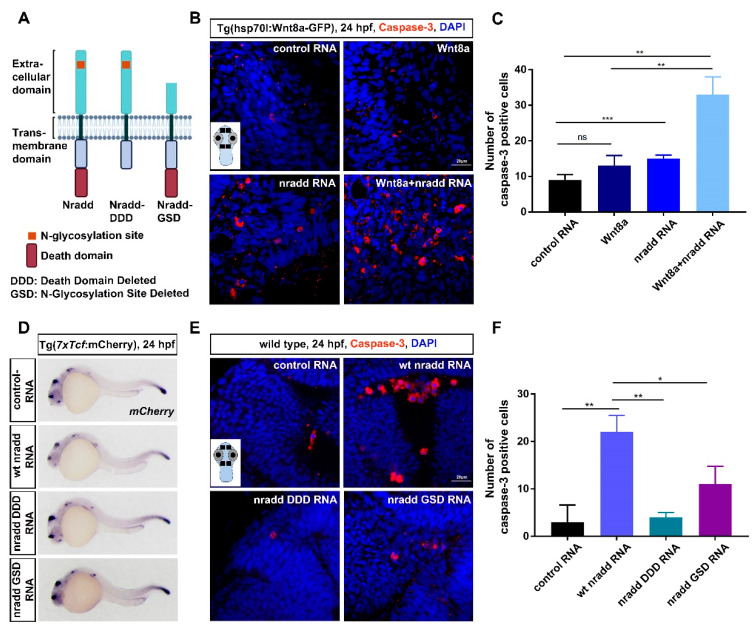
Nradd acts together with Wnt/β-catenin signaling to promote apoptosis during development. (**A**) Schematic representation shows domain structures of wt Nradd, Nradd without the death domain (Nradd death domain deleted (DDD)) and Nradd without the N-glycosylated N-terminal region (Nradd glycosylation site deleted (GSD)). (**B**) Anti-cleaved caspase 3 staining of control, Wnt8a activated, *nradd* mRNA-injected and Wnt8a activated+*nradd* mRNA-injected zebrafish embryos at 24 hpf. Sections are counterstained for DAPI. The inset shows a representative embryo with four different regions indicated with black rectangles that are used for counting the cleaved caspase 3-positive cells. The embryo image is created with BioRender.com. Scale bar: 20 μm. *nradd* mRNA (250 pg) induces apoptosis in zebrafish embryos compared to control group and also acts together with Wnt8a to further enhance apoptosis. The experiment was performed in hs:Wnt8 transgenic embryos where Wnt8a expression is induced by giving a heat shock at 60% epiboly. Four independent experiments were conducted. (**C**) Quantification of cleaved caspase 3-positive apoptotic cells shown in (B). Numbers represent average number of apoptotic cells counted from four different regions of 20 embryos for each group. Error bars are SD, ** indicates *p* < 0.01 and *** indicates *p* < 0.001. (**D**) WMISH showing the loss of capacity in *nradd DDD* (250 pg) and *nradd GSD* (250 pg) mRNA to inhibit canonical Wnt signaling. mCherry WMISH shows downregulation of signaling in the transgenic 7xTcf:mCherry Wnt/β-catenin reporter embryos by wt *nradd* (58/61). Overexpression of *nradd DDD* (44/46) or *nradd GSD* (51/52) cannot inhibit Wnt/β-catenin signaling. Three independent experiments were conducted. (**E**) Anti-cleaved caspase 3 staining of control (100 pg), wt *nradd* (250 pg), *nradd DDD* (250 pg) and *nradd GSD* (250 pg) mRNA-injected zebrafish embryos at 24 hpf. Sections are counterstained for DAPI. The inset shows a representative embryo with four different regions indicated with black rectangles that are used for counting the cleaved caspase 3-positive cells. Scale bar: 20 μm. *nradd* mRNA (250 pg) induces apoptosis in zebrafish embryos compared to control group, while *nradd DDD* or *nradd GSD* cannot. Three independent experiments were conducted. (**F**) Quantification of cleaved caspase 3-positive apoptotic cells shown in (**E**). Numbers represent average number of apoptotic cells counted from four different regions of 20 embryos for each group. Error bars are SD, * indicates *p* < 0.05 and ** indicates *p* < 0.01.

**Figure 6 biomolecules-11-00100-f006:**
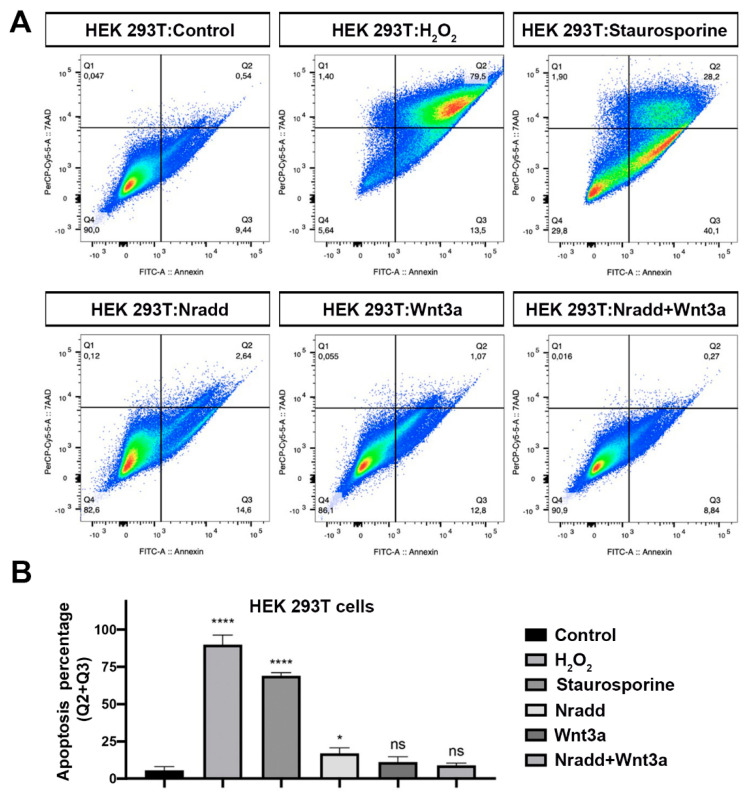
Zebrafish Nradd promotes apoptosis in human embryonic kidney cells. (**A**) Apoptosis assay on HEK293T cells using flow cytometry after staining with annexin V-FITC and 7AAD. Representative scatter plots of 7AAD (y-axis) vs. annexin V-FITC (x-axis). Nradd significantly enhances apoptosis. (**B**) Percentage of apoptotic cells, * indicates *p* < 0.05, **** is *p* < 0.0001 and ns is non-significant. All experiments were performed in five replicates. Three independent experiments were conducted.

**Figure 7 biomolecules-11-00100-f007:**
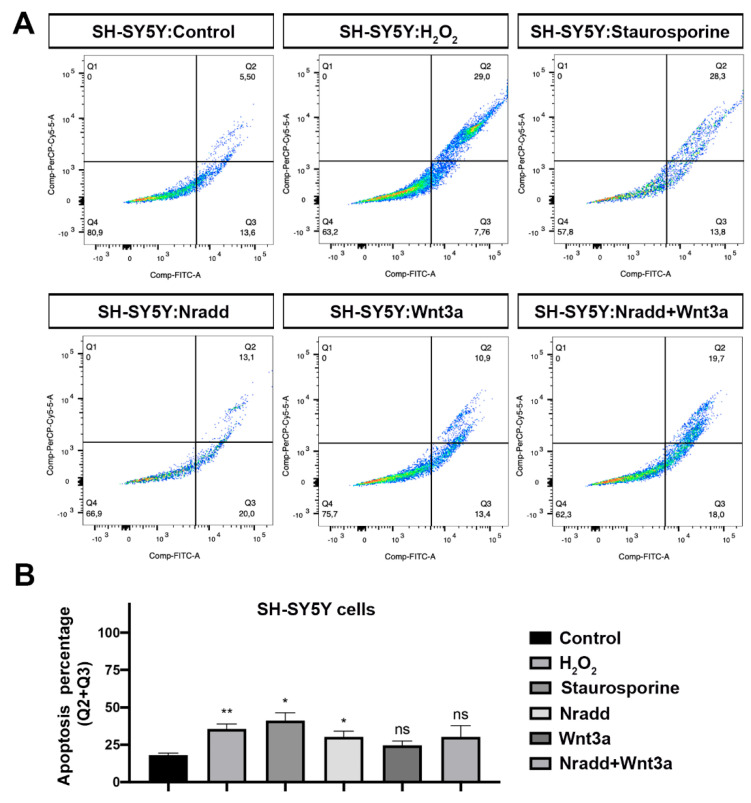
Zebrafish Nradd enhances apoptosis in neuroblastoma cells. (**A**) Apoptosis assay on SH-SY5Y cells using flow cytometry after staining with annexin V-FITC and 7AAD. Representative scatter plots of 7AAD (y-axis) vs. annexin V-FITC (x-axis). Nradd significantly induces apoptosis. (**B**) Percentage of apoptotic cells, * indicates *p* < 0.05, ** is *p* < 0.01 and ns is non-significant. All experiments were performed in five replicates. Three independent experiments were conducted.

## Data Availability

Not applicable.

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
