# Peer review of "Nradd Acts as a Negative Feedback Regulator of Wnt/β-Catenin Signaling and Promotes Apoptosis"

_biomolecules, 2021, doi:10.3390/biom11010100_

Round 1

Reviewer 1 Report

The authors report identification of zebrafish Nradd (zNradd), the homologue of human p57NTR, to negatively regulate Wnt/b-catenin signaling in zebrafish and human cell lines. Further, Nradd affects Wnt8 mediated mesodermal and neuroectodermal patterning in zebrafish embryos. The authors show localization of overexpressed Nradd at the plasma membrane and suggest a role at the receptor level through interaction with Lrp6. The role in apoptosis and localization to the plasma membrane was known for other homologs and model organisms before and is shown for zNradd in this manuscript. Therefore the most intriguing aspect is the role in Wnt signaling. However, the mechanism could have been investigated in more detail.

The manuscript is well written and of relatively broad interest of readers to the fields of cell signaling, developmental biology and cell death pathways. Figures are of good quality, with very few exceptions.

Major comments:

However, many controls are missing, sometimes experimental number and the setup (immmunoprecipitation) has to be improved. This needs to be addressed experimentally.

One concern is that authors mostly use overexpression of Nradd in zebrafish and plasmid transfection into the human system. This should be expanded by knockdown studies e.g. by morpholinos in zebrafish (in conditions of Wnt-on mode) as rescue experiments to rule out overexpression artifacts.

It remains unclear why zNradd was tested in human cells when it is a pseudogene in human cells. Therefore, the role of the homolog p75NTR should be tested in Wnt signaling rather than overexpressing zNradd.

This would help to further improve the quality of the manuscript.

Figure 1B: Positive controls should be added to show Axin1 induction by heat shock and Wnt8 at protein level.

Figure 2D: Effects on axin2 are rather minor. Why does Wnt8 result in such a mild effect? How can the authors rule out suboptimal activation of Wnt signaling?

Figure 2G-H: Expression controls need to be added to show expression of Wnt8 and Nradd in the cells.

Figure 3A: Expression controls for nradd have to be added

Figure 4A: Nradd-GFP should be shown also in absence of CellMask-Deep Red. How to be sure that the signal does not affect the other channel?

Figure 4D: Negative controls need to be shown. Lrp6 has the tendency to unspecifically bind to proteins. The authors should show that it does not bind to the beads or GFP alone. Also, the quality of the blot should be improved. The immunprecipitation experiment should be performed from zebrafish lysates to show interaction in the relevant model.

Figure 5: Expression of wt Nradd and the mutants needs to be shown. One important aspect, which has to be considered when monitoring deletion mutants and negative effects of them is: Are the mutants expressed and stable at protein level? Are the protein levels comparable to wt?

Figure 5D: Authors should evaluate the effect by statistical analysis.

Figure 5: Was cleaved caspase counted in selected sections? How were the sections chosen? Cleaved caspase should be investigated by an additional (unbiased) method, for example western blot.

Figure 6 and 7:

nradd needs to be shown in cells. Wnt3a functionality needs to be shown when used in the AnnexinV assay.

Line 172 & Line 196: Was only one technical triplicate performed in qPCRs and reporter assays? The term “three samples” might be misleading. 3 independent experiments measured in technical triplicates have to be analyzed throughout the manuscript. This should be clearly indicated throughout the manuscript (as in Figure 6,7). E.g. how often were WMISH performed independently?

Line 334:

Negative feedback has not been shown in human cells as Nradd is a pseudogene there. The conclusion should be rephrased. What about the fact that Nradd is a pseudogene in humans? Does p75NTR take over? What about the role of p75NTR in zebrafish? See also line 468 (comment below).

Line 371:

How can one rule out that this is not merely an effect of overexpression (and an artifact)? To further support the conclusion of the authors, knockdown studies using morpholinos should be performed and rescue assays to show specificity of the observed effects. This would help to further improve the quality of the manuscript.

Line 441: This remains unclear: How was this demonstrated to be a “synergistic" effect? Is the effect more than additive?

Line 468: Are there publications available about the pseudogene inactivation?

Why is zNradd tested in human cells at all when it is a pseudogene in humans with no obvious function? Does the presence of Nradd convey advantages that are exclusively found in zebrafish then? Which other family members would be taking over and compensate for absence of Nradd in human cells? The authors should test the role of p75NTR in Wnt signaling in human cells.

Line 522: Experimental data would help to support this assumption: Does interfering with Wnt signaling inhibit apoptosis then?

Line 544: This has not been demonstrated experimentally. A late apoptotic event should be tested experimentally in order to exclude the role in late apoptosis

Line 158: Were the PCR products not verified by sequencing then? The generation of all constructs is only indicated by restriction digestion. Sequencing is essential.

What about the induction of human Nradd homologs in response to Wnt? Are those also negative feedback regulators? Or is this mechanism of the Nradd family being regulated by Wnt not present in the human system?

Minor comments:

Line 74: Please, add references

Line 172: Since two reference genes were used for normalization, how was calculation performed?

Line 200: The authors should indicate machines used for measuring the reporter assay.

Line 201: Why do authors sometimes show SEM and sometimes SD?

Line 332: The pBAR reporter should be introduced a bit more here

Line 572: This remains unclear: If Nradd is a pseudogene in humans, why and how could it be drugged then? Or do authors suggest overexpressing Nradd in humans?

What about other family members is there a known role of one of them in Wnt signaling?

The authors should comment why these particular 3 cell lines were chosen

Some references are not aligned properly. See for example reference #13, #24…

Reviewer 2 Report

It is a well written manuscript. However, the quality of western blot images (IP and input) is poor. The authors must replace them with better quality images.

Reviewer 3 Report

The manuscript by Ozalp et al. presents a comprehensive study on the role and regulation of Nradd by the WNT/b-catenin signaling pathway. The data are technically sound and valid, however, certain concerns must be addressed.

  1. The Authors used heat shock to modulate activity of WNT/b-catenin signaling (section 3.1) and to select target genes of this pathway. Is this specific way of modulating this particular signaling pathway in the experimental model that was used in this study? Heat shock can induce multiple responses in cells.
  2. Statistical analysis for Fig. 2 should be performed again. Eg. panels D-F - Anova should be used to compare three groups. For panel G and H it can be concluded that nradd DOES NOT significantly inhibit WNT8/3a CM-induced WNT/b-catenin pathway activity. In addition, Y axis in panels G and H (fold change) should be more clearly labeled - fold change in what?
  3. Fig. 5C, F - please change Y axis to: number of caspase-3-positive cells.
  4. Fig. 6B - please specify which cells are considered as apoptotic? Total % of Annexin-V-positive? Please specify Y axis label accordingly.

Round 2

Reviewer 1 Report

The authors have addressed almost all points by textual clarifications and have made appropriate changes in the manuscript.   The answer for point #24 is still missing, this is only a minor point, which the authors still could comment now.   My answer to their Response #10,  I am still not convinced with the result in Figure 4D, as the negative control is missing.  However, since the point to point response will be published with the paper anyway and will be accessible to all readers,  so I feel we can leave the Figure as it is.     Taken together, I think the paper can be accepted for publication.